# Comparisons of Hematological and Biochemical Profiles in Brahman and Yunling Cattle

**DOI:** 10.3390/ani12141813

**Published:** 2022-07-15

**Authors:** Yu Yang, Shuling Yang, Jia Tang, Gang Ren, Jiafei Shen, Bizhi Huang, Chuzhao Lei, Hong Chen, Kaixing Qu

**Affiliations:** 1Key Laboratory of Animal Genetics, Breeding and Reproduction of Shaanxi Province, College of Animal Science and Technology, Northwest A&F University, Yangling, Xianyang 712100, China; yangyukate@163.com (Y.Y.); yangshuling@nwafu.edu.cn (S.Y.); candy951118@gmail.com (J.T.); rengang666@nwafu.edu.cn (G.R.); shenjiafei0118@163.com (J.S.); leichuzhao1118@126.com (C.L.); 2Academy of Science and Technology, Chuxiong Normal University, Chuxiong 675000, China; 3Yunnan Academy of Grassland and Animal Science, Kunming 650212, China; hbz@ynbp.cn; 4College of Animal Science, Xinjiang Agricultural University, Urumqi 830052, China

**Keywords:** Yunling cattle, Brahman, hematology, blood biochemical index

## Abstract

**Simple Summary:**

In order to evaluate the health condition and differences between Brahman cattle and Yunling cattle, 28 adult Brahman and 65 adult Yunling cattle were analyzed from the same farm with standard grazing management. We detected 55 hematological and biochemical parameters using an automatic biochemical analyzer. The results showed that 27 hematological and biochemical parameters of Brahman cattle were lower than those of Yunling cattle, whereas the other indices detected were higher. Based on the hematological and biochemical parameters investigated, the Yunling cattle may have a better physical condition than Brahman cattle, such as having a greater metabolism and better liver function. Thus, these data suggest that, as a hybrid breed that is half Brahman, Yunling cattle are superior in their abilities of adaptability, stress resistance and tolerance to crude feed. Therefore, our study reveals the underlying molecular basis of Yunling cattle’s better performance at adapting to local environments, and maintaining beef production and reproduction traits.

**Abstract:**

Brahman cattle are tolerant to parasite challenges and heat stress. Yunling cattle are three-way hybrids that are half Brahman cattle, a fourth Murray Grey cattle and a fourth Yunnan Yellow cattle, with good beef performance. The hematological and biochemical parameters can reflect the physiology and metabolic conditions of cattle, and there are valuable indicators of production performance and adaptability that can be found by studying the cattle. To assess the health status and differences, we compared 55 hematological and biochemical parameters of 28 Brahman cattle and 65 Yunling cattle using an automatic biochemical analyzer. Our results showed that 27 hematological and biochemical indices of Brahman cattle were lower than those of Yunling cattle, whereas the other parameters were higher. There are 20 indices with significant differences that were detected between Brahman and Yunling cattle (with *p* ≤ 0.01 or 0.01 ≤ *p* ≤ 0.05, respectively), and no significant differences were found for other indices (*p* > 0.05). Based on these results, Yunling cattle may have a better physical condition than Brahman cattle, may be better at adapting to local environments, and can maintain a good production and reproduction performance. As a new breed that is half Brahman, the abilities of Yunling cattle, including adaptability, stress resistance and tolerance to crude feed, were better than Brahman cattle under the same management conditions.

## 1. Introduction

The Brahman (American zebu) was cultivated in America, and can tolerate parasite challenges and heat stress [1]. Additionally, tolerance to rough feeding, adaptability to environmental conditions, the ability to forage, good production performance under barren and arid conditions, and the ability to reproduce in extreme high temperatures are traits of the Brahman, which made it contribute significantly to the cattle industry both in America and China [2,3]. Yunling cattle were bred to be resistant to heat, to produce a high meat yield and to be suitable to the local climate in Yunnan province, China, which is located in a subtropical zone. Yunling cattle are a composite breed, created by three-way crossbreeding, that are half Brahman, a fourth Murray Grey, and a fourth Yunnan Yellow cattle, which was achieved and developed in Yunnan, China [4]. Although some studies have shown that Yunling cattle have a good fattening performance, large body size, high meat production, good carcass traits and fatty acid profiles in meat, the detail mechanisms are still unknown [5]. Thus, further investigations are needed in order to better understand the basis of the formation of these good characteristics in Yunling cattle.

Blood indicators can reflect the health conditions of cattle: some diseases such as anemia, abnormal liver and kidney function can be diagnosed at an early stage and appropriate treatment can be given in time. Blood is an opaque red liquid flowing in the heart and blood vessels, and its main components are plasma, blood cells and platelets. Blood is responsible for transporting oxygen and various nutrients to various tissues and organs, transporting wastes and metabolites produced by organs to corresponding organs for decomposition. At the same time, it also plays an important role in the regulation of body temperature, the maintenance of acid-base balance in the body, the regulation of organ activity and the immunity of animals. It is indispensable for the normal function of animals. When a physiological and pathological change occurs in animal, the blood components change accordingly. Therefore, the blood parameters can be clinically used to judge the disease condition and effectively apply different treatments.

At present, there have been many studies utilizing the hematological and biochemical parameters for animal breeding selection [6,7,8,9,10]. Hematological and biochemical profiles are one of the most sensitive indicators that reflect the physiology, health and metabolic conditions of cattle, as they have important reference values on the production performance and adaptability of cattle. Therefore, in this study the blood of Brahman and Yunling cattle was collected, and the comparison of 55 hematological and biochemical parameters was performed. Our study provides some findings on the blood indicators of the differential production performance of Brahman and Yunling cattle, may regard health levels, feeding and breeding, and disease prevention.

## 2. Materials and Methods

### 2.1. Animals and Management

The breed processes were conducted on Xiaoshao Farm, Yunnan Academy of Grassland and Animal Science (25°22′ N and 103°04′ E). The farm had a 24 h free-grazing system. All cattle were female and >4 years old, and had no unhealthy clinical manifestations within 30 days before blood sampling.

### 2.2. Blood Collection and Detection

The experimental cattle were taken from the free-grazing farm in the morning. After disinfecting the skin of the blood collection site with 75% alcohol, 5 mL, 3–4 mL and 1–3 mL of whole blood samples from the jugular vein were taken into aseptic tubes containing coagulant, lithium heparin and EDTA, respectively. All the samples were stored in foam boxes with ice packs or vaccine storage boxes. The interval between the first and the last blood collection did not exceed 2 h, and the blood was sent to the laboratory within 4 h.

The blood samples of 28 Brahman and 65 Yunling cattle were analyzed by using an automatic biochemical analyzer to test for 55 hematological and biochemical profiles (automatic biochemical analyzer; manufacturer: Sysmex Co., Ltd. of Japan; model: XB-2100).

### 2.3. Parameters Detected

White blood cells (WBCs), percentage of monocytes (MONO%), percentage of basophils (BAS%), number of monocytes (MONO^#^), number of basophilic granulocytes (BAS^#^), erythrocytes (RBCs), hemoglobin (HGB), hematocrit (HCT), mean corpuscular volume (MCV), mean corpuscular hemoglobin (MCH), mean corpuscular hemoglobin concentration (MCHC), red cell volume distribution width-CV (RDW-CV%), red cell volume distribution width-SD (RDW-SD), absolute value of reticulocytes (RET^#^), platelets (PLTs), thrombocytocrit (PCT%), mean platelet volume (MPV), platelet distribution width (PDW%), platelet large cell ratio (PLCR%), percentage of reticulocytes (RET%), triglyceride (TG), total cholesterol (T-CHOL), high-density lipoprotein cholesterol (HDL-C), low-density lipoprotein cholesterol (LDL-C), total bilirubin (STB,TBIL), direct bilirubin (DBIL), indirect bilirubin (IBIL), total protein (TP), albumin (ALB), globulin (GLO), ALB:GLO, prealbumin (PAB), aspartate aminotransferase (AST), alanine aminotransferase (ALT), AST:ALT, glutamyl transpeptidase (GGT), alkaline phosphatase (ALP), cholinesterase (CHE), total bile acid (TBA), glucose (GLU), urea nitrogen (BUN), uric acid (URIC), creatinine (CREA), HCO3-, plasma kalium (K^+^), plasma natrium (Na^+^), plasma chlorine (Cl^−^), plasma calcium (Ca^2+^), plasma phosphorus (P^5+^), plasma magnesium (Mg^2+^), homocysteine (HCY), creatine kinase (CK), creatine kinase isoenzymes (CKMBs), lactate dehydrogenase (LDH), hydroxybutyrate dehydrogenase (HBDH).

### 2.4. Statistical Analyses

All data were presented as mean ± standard deviation ($\bar{x}$ ± SD). A *t*-test was employed to analyze the differences between Brahman and Yunling cattle for independent samples using the SPSS 18.0 software. The distribution of variables was assessed for normality with histograms that are showed in the Appendix A.

## 3. Results

### 3.1. The Results of Hematology

The Brahman cattle had lower values than Yunling cattle in 11 hematological parameters, of which 3 showed significant differences, and 2 showed extremely significant differences; however, Brahman cattle had higher values than Yunling cattle in the other 9 parameters (Table 1). The WBC, MONO%, BAS%, MONO ^#^ and BAS ^#^ levels of Brahman were lower than Yunling cattle. Among these indicators related to WBCs, the BAS^#^ of Brahman was 0.0258 and of Yunling cattle was 0.0311, which was significant (*p* = 0.032) and BAS was significant (*p* = 0.047). The RBC, HGB, HCT, MCV, MCH and RET levels of Brahman were lower than Yunling cattle, but MCHC, RDW-CV%, RDW-SD were to the contrary. Among these indicators related to RBCs, the RBC (*p* = 0.037), HGB (*p* = 0.003) and HCT (*p* = 0.002) levels of Brahman were significantly or extremely significantly lower than Yunling cattle. The PLTs, PCT%, MPV, PDW%, PLCR% and RET% of Brahman were higher values than those of Yunling cattle. Among these indicators, those related to PLTs, MPV (*p* = 0.027) and PDW% (*p* = 0.028) of Brahman showed significantly higher levels than those of Yunling cattle. All significant results of hematology are displayed in Table 2.

### 3.2. The Comparisons of Blood Biochemical Indicators

As shown in Table 3, 16 of the 35 blood biochemical indicators in Brahman cattle had lower values than Yunling cattle, and others were higher. There were six parameters that had significant differences between these two cattle breeds, and six parameters with extremely significant differences. The TG, DBIL, TP, ALB, ALB:GLO, PAB, ALT, CHE, TBA, BUN, CREA, HCY and CKMB levels of Brahman were not significantly lower than those of Yunling cattle. The T-CHOL, HDL-C, LDL-C, GLO, AST, AST: ALT, ALP, HCO^3−^, K^+^, Na^+^, Mg^2+^, CK, LDH and HBDH levels of Brahman were not significantly higher than those of Yunling cattle. The ALB:GLO (*p* = 0.029) ratio of Brahman was significantly higher than that of Yunling cattle. The TBIL (*p* = 0.039) of Brahman showed a significantly lower level than in Yunling cattle. The IBIL (*p* = 0.002), ALB (*p* < 0.001), GGT (*p* = 0.001) and Ca^2+^ (*p* = 0.006) of Brahman showed extremely significantly lower levels than in Yunling cattle. The HDL-C (*p* = 0.01), URIC (*p* = 0.015), Cl^−^ (*p* = 0.027) and P^5+^ (*p* = 0.028) of Brahman showed extremely significantly higher levels than in Yunling cattle (Table 4).

## 4. Discussion

Hematological and biochemical parameters are not only the main basis for the diagnosis of various blood diseases, but also provide important information for the diagnosis and identification of other diseases, which can reflect the health status of the body [11,12,13,14].

In this study, Brahman cattle and Yunling cattle were kept in Xiaoshao pasture with an altitude of 1910 m and an average temperature of about 15 °C, and were used to explore the adaptation of both to the local environment.

WBCs, RBCs and PLTs are important contents in the blood. WBC counts and their differential are established systemic inflammatory markers. WBCs is a collective including neutrophils, lymphocytes, monocytes, eosinophils and basophils. As a part of the immune system, WBCs provide the body with the ability to resist diseases and harmful substances in the external environment, and participate in the body’s immune response. Yunling cattle have significantly higher levels of BAS% and BAS^#^ than Brahman cattle. RBCs are the most abundant components in blood, and their main functions are to carry oxygen, act as respiratory carriers of carbon dioxide, maintain acid-base balance, etc. [15]. HGB has a similar effect as RBCs. HCT is the ratio of the volume of blood cells in the blood [16]. Within the normal range, the larger the HCT level, the stronger the blood oxygen delivery capacity. Reticulocytes can reflect the production of bone marrow erythrocytes. Yunling cattle, with a significantly higher RBC level and extremely significantly higher HGB and HCT levels, may have better adaptability to high altitudes than Brahman cattle. Better adaptability to high altitudes suggests that Yunling cattle, which are bred in the Yunnan province of China, may be more suitable for feeding in different climates. PLTs are the main cellular mediator of hemostasis, which include the platelet count and platelet-related parameters [17]. Recently, a rapidly expanding amount of research has uncovered the role of PLTs in development, infection, inflammation, inflammatory hemostasis, organ repair, and even cancer and sepsis [18]. Yunling cattle have lower levels of PLTs, PCT%, PLCR% and RET%, with significantly lower levels of MPV and PDW%. These may suggest that Brahman cattle have a better recovery effect than Yunling cattle in local or small-scale trauma.

TG, T-CHOL, HDL-C and LDL-C are usually used as indicators of blood lipid testing. Hyperlipidemia could cause arterial plaque and accelerate arteriosclerosis, and arterial diseases are more likely to occur after middle age. LDL-C is traveled from the liver to tissues throughout the body, and HDL-C is traveled from each tissue back to the liver for metabolism. In our results, the HDL-C of Brahman cattle was significantly higher than that of Yunling cattle. The ratio of HDL-C/LDL-C of Yunling cattle was 6.98, and the ratio of HDL-C/LDL-C of Brahman cattle was 7.35, which is much higher than that of Yunling cattle. High HDL-C and HDL-C/LDL-C affect the deposition of fat in peripheral tissues, which may be the main reason why it is more difficult for Brahman cattle to produce high-grade beef compared to Yunling cattle [19,20,21]. The conclusion is consistent with the previous results that Brahman cattle are not suitable for producing marbled beef.

The bilirubin in the serum is mainly derived from the HGB produced by the destruction of senescent RBCs. DBIL undergoes glucuronidation in the liver, and IBIL does not. STB is the sum of DBIL and IBIL. STB (*p* = 0.039), IBIL (*p* = 0.002) and DBIL of Brahman all showed lower levels than Yunling cattle. On the one hand, the RBCs of Yunling cattle are significantly higher than those of Brahman, which leads to a significant or extremely significant difference in STB and IBIL between the two breeds. On the other hand, it may suggest that the metabolism of Yunling cattle is greater than that of Brahman, and the ability of Yunling cattle’s HGB to perform glucuronidation in the liver is better than that of Brahman cattle.

Liver function is very important for cattle. Improper storage of feed can make feed moldy, and moldy feed can seriously damage the liver. When the bovine body suffers from other diseases, bad liver function tends to aggravate the condition. TP, ALB, GLO, AST, ALT, GGT and ALP are indicators of liver function detection, which directly reflect the metabolic and reserve capacity of the liver. In our studies, the ALB (*p* < 0.001) of Brahman showed extremely significantly lower levels than Yunling cattle, and ALB:GLO (*p* = 0.029) was significantly lower, as it should be. ALB can affect the existence of many ligands in the blood circulation. Besides the well-known act of expanding plasma, ALB, which is synthesized by the liver, has the function of immuno-modulation, binds and transports many endogenous and exogenous substances, causes anti-inflammatory activity, and so on [22,23]. Hypoalbuminemia caused by low ALB can be related to many different diseases, including liver cirrhosis, nephrotic syndrome, malnutrition and sepsis [24]. Therefore, the higher ALB in the healthy range means Yunling cattle have better liver function, steady immunity and better nutrient delivery than Brahman cattle [25]. GGT (*p* < 0.001) of Brahman showed extremely significantly higher levels than Yunling cattle. GGT is widely distributed in various tissues, but is mostly in the kidney, followed by the pancreas and liver. Elevated GGT is commonly found in many liver diseases, and patients with liver cirrhosis, hepatitis, and intra-hepatic and extra-hepatic bile duct obstruction always have high GGT levels [26,27]. The higher GGT of Brahman cattle were identified, showing that they may have higher rates of hepatitis. The results support the same conclusion, that Yunling cattle may have better liver function than Brahman cattle.

Many studies suggested that GLU is related to metabolism and diet [28,29]. GLU can reflect the balance between sugar production and tissue consumption in the body. Bailey’s study showed that the provision of supplemental GLU reduces forage intake and digestibility [30]. Extremely significantly higher levels of GLU in Yunling cattle may mean better feed utilization and energy storage for cattle. Their work also suggested that acts of self-control are caused by blood GLU levels. That may explain the “unstable and uncontrollable mood” which Brahman show daily as compared to Yunling cattle.

BUN, URIC and CREA can reflect the three aspects of kidney function. Chikhou et al. [31] showed that the level of BUN in plasma could reflect the protein metabolism of animals and could be used as an indicator of the body’s protein decomposition status. Under the condition of a low nitrogen diet, ruminants mainly increased nitrogen deposition through the BUN cycle; thus, the content of BUN increases to meet the needs of the body [32]. Compared with Brahman, Yunling cattle have higher BUN and lower URIC levels, indicating that Yunling cattle have a better roughage tolerance than Brahman, which is consistent with the actual production. Crea is a product of the muscle and phosphate metabolism in animal bodies. The concentration of Crea in bovine blood is mainly affected by skeletal muscle content, exercise capacity and renal function. The Crea of Yunling is higher than that of Brahman cattle, which may be due to Yunling cattle having a stronger exercise capacity and muscle metabolism than Brahman cattle.

K^+^, Na^+^, Cl^−^, Ca^2+^, P^5+^ and Mg^2+^ are electrolytes in the body, and HCO^3−^ is a buffer to maintain the body’s pH level. The concentration of plasma inorganic ions plays an important role in judging the osmotic pressure in the body, blood pH, muscle excitability, and heart and kidney function. The difference in these may be because of different cattle breeds.

CK, CKMB and LDH are the contents of the myocardial enzyme spectrum. CK is mainly found in skeletal and heart muscles. In muscle atrophy caused by skeletal muscle diseases, CK activity increases. Differences in these values may be related to muscle.

## 5. Conclusions

In physiology, hematological and biochemical indices provide useful information for animal health. These parameters detected in our study can be used as phenotypic parameters for Yunling cattle and Brahman cattle to adapt to the natural environment in Kunming, China, as they describe the basic hematological and biochemical indicators in Yunling and Brahman cattle.

Hematology and blood biochemical parameters can reflect the dynamic interplays between animals and the environment. Some indicators are significantly different between Yunling and Brahman cattle. Under the same condition, as a new beef breed containing half the bloodline of Brahman cattle, Yunling cattle might show better physical conditions, such as having greater metabolism and better liver function. The traits are speculative and must be confirmed in further studies with actual studies testing these hypotheses specifically. Our data suggest that Yunling cattle can better adapt to the local environment than Brahman cattle in Kunming, China. In other words, the Yunling cattle show good adaptability, stress resistance and tolerance to rough feeding, and can maintain good production and reproduction performance in the special environment of Kunming.

## Figures and Tables

**Table 1 animals-12-01813-t001:** The hematology indicators of Brahman and Yunling cattle.

Item	Yunling Cattle		Brahman	
	Mean ± SD	Range	Mean ± SD	Range
WBCs (×10^−9^/L)	9.78 ± 2.53	5.85–15.63	9.29 ± 2.29	4.95–14.44
MONO%	7.13 ± 2.06	2.7–12.7	6.74 ± 1.88	3.4–11.2
BAS%	0.33 ± 0.14 ^a^	0.1–0.7	0.26 ± 0.12 ^b^	0.1–1
MONO^#^	0.66 ± 0.21	0.3–1.34	0.62 ± 0.17	0.28–0.91
BAS^#^	0.0311 ± 0.0134 ^a^	0.01–0.07	0.0258 ± 0.0090 ^b^	0.01–0.05
RBCs (×10^−12^/L)	8.41 ± 0.94 ^a^	7.16–11.36	7.94 ± 0.99 ^b^	5.18–9.55
HGB (g/L)	150.39 ± 13.76 ^A^	122–194	140.39 ± 14.29 ^B^	112–166
HCT (L/L)	0.41 ± 0.04 ^A^	0.35–0.5	0.38 ± 0.05 ^B^	0.31–0.48
MCV (fL)	48.87 ± 4.55	37–57.9	48.40 ± 6.72	37.9–64.1
MCH (pg)	17.95 ± 1.28	14.3–20.6	17.64 ± 1.41	15–20.5
MCHC (g/L)	368.40 ± 15.49	330–401	372.48 ± 17.77	337–400
RDW-CV%	23.13 ± 1.84	20.3–29	23.65 ± 2.33	18–29
RDW-SD (fL)	36.40 ± 2.66	30.7–41.4	36.01 ± 3.43	31.6–44.4
RET (10^−12^/L)	0.0067 ± 0.0017	0–0.01	0.0076 ± 0.0023	0–0.01
PLTs (10^−9^/L)	258.43 ± 99.98	33–475	266.62 ± 98.65	8–487
PCT%	0.22 ± 0.06	0.06–0.35	0.24 ± 0.07	0.14–0.42
MPV (fL)	7.94 ± 0.76 ^b^	6.6–9.8	8.36 ± 0.64 ^a^	7–9.7
PDW%	9.06 ± 1.30 ^a^	6.9–11.5	9.78 ± 1.10 ^b^	7.5–11.8
P-LCR%	12.53 ± 5.07 ^a^	4.4–25.1	15.15 ± 4.50 ^b^	6.1–24.4
RET%	0.0827 ± 0.02	0.05–0.14	0.0861 ± 0.02	0.06–0.14

Note: Values without common superscript letters in the same row differ at *p* ≤ 0.05 for lower-case and *p* ≤ 0.01 for upper case.

**Table 2 animals-12-01813-t002:** Comparisons of hematology indicators between Brahman and Yunling cattle.

Item	Yunling Cattle		Brahman		*p*-Value
	Mean ± SD	*n*	Mean ± SD	*n*	
BAS (%)	0.33 ± 0.14 ^a^	65	0.26 ± 0.12 ^b^	25	0.047
BAS^#^ (10^−9^/L)	0.0311 ± 0.0134 ^a^	64	0.0258 ± 0.0090 ^b^	26	0.032
RBCs (10^−12^/L)	8.41 ± 0.94 ^a^	65	7.94 ± 0.99 ^b^	26	0.037
HGB (g/L)	150.39 ± 13.76 ^A^	65	140.39 ± 14.29 ^B^	26	0.003
HCT (L/L)	0.41 ± 0.04 ^A^	65	0.38 ± 0.05 ^B^	26	0.002
MPV (fL)	7.94 ± 0.76 ^b^	54	8.36 ± 0.64 ^a^	21	0.027
PDW%	9.06 ± 1.30 ^a^	54	9.78 ± 1.10 ^b^	21	0.028
P-LCR%	12.53 ± 5.07 ^a^	53	15.15 ± 4.50 ^b^	21	0.042

Note: Values without common superscript letters in the same row differ at *p* ≤ 0.05 for lower-case and *p* ≤ 0.01 for upper case.

**Table 3 animals-12-01813-t003:** Comparisons of blood biochemical indicators between Brahman and Yunling cattle.

Item	Yunling Cattle		Brahman	
	Mean ± SD	Range	Mean ± SD	Range
TG (μmol/L)	0.24 ± 0.09	0.11–0.41	0.23 ± 0.07	0.13–0.37
T-CHOL (mmol/L)	3.24 ± 0.79	1.97–5.36	3.48 ± 0.81	2.15–5.13
HDL-C (mmol/L)	2.14 ± 0.37 ^B^	1.44–3.11	2.37 ± 0.39 ^A^	1.63–3.04
LDL-C (mmol/L)	0.3069 ± 0.1557	0.02–0.8	0.3221 ± 0.1360	0.13–0.59
TBIL (μmol/L)	2.85 ± 0.99 ^a^	1–5.3	2.44 ± 0.74 ^b^	1.2–4.1
DBIL (μmol/L)	1.13 ± 0.40	0.4–2.1	1.10 ± 0.33	0.3–1.9
IBIL (μmol/L)	1.74 ± 0.75 ^A^	0.4–3.4	1.33 ± 0.50 ^B^	0.4–2.3
TP (g/L)	81.17 ± 7.48	65.4–95.4	81.41 ± 6.17	71.2–95.2
ALB (g/L)	36.43 ± 2.01 ^A^	31.9–42.4	33.63 ± 3.15 ^B^	26.7–38.4
GLO (g/L)	44.73 ± 7.59	30.8–58.6	46.58 ± 9.17	21.2–67.5
ALB:GLO	0.84 ± 0.16 ^a^	0.61–1.21	0.76 ± 0.19 ^b^	0.41–1.26
PAB (mg/L)	38.57 ± 4.93	29–49	39.67 ± 4.69	32–48
AST (U/L)	83.79 ± 33.07	35–195	93.96 ± 29.93	51–160
ALT (U/L)	28.42 ± 8.05	13–50	27.68 ± 6.83	12–43
AST:ALT	3.00 ± 1.09	1.36–6.59	3.33 ± 0.84	2–5.5
GGT (U/L)	19.51 ± 9.42 ^B^	3–57	32.04 ± 15.82 ^A^	1–67
ALP (U/L)	90.97 ± 43.40 ^B^	30–220	114.43 ± 39.55 ^A^	39–224
CHE (U/L)	242.19 ± 58.10	120–354	238.93 ± 38.72	161–302
TBA (μmol/L)	18.45 ± 10.26	4.7–42.2	14.82 ± 8.81	3.9–39.5
GLU (mmol/L)	4.28 ± 0.49 ^A^	3.56–6.18	3.80 ± 0.82 ^B^	2.54–5.9
BUN (mmol/L)	4.99 ± 1.78	2.16–8.67	4.82 ± 1.24	2.53–7.64
URIC (μmol/L)	39.86 ± 14.25 ^b^	16–82	47.75 ± 13.81 ^a^	19–71
CREA (μmol/L)	123.03 ± 32.97	77–218	112.85 ± 27.39	72–178
HCO_3_^−^ (mmol/L)	25.83 ± 2.62	18.6–32.3	26.34 ± 3.30	17.4–30.3
K^+^ (mmol/L)	4.12 ± 0.46	3.17–5.28	4.25 ± 0.65	2.22–5.86
Na^+^ (mmol/L)	141.23 ± 3.46	133.1–149	142.36 ± 2.17	136.6–148.3
Cl^−^ (mmol/L)	98.32 ± 3.21 ^b^	88–103.9	100.06 ± 3.94 ^a^	88.6–106.9
Ca^2+^ (mmol/L)	2.38 ± 0.11 ^A^	2.08–2.63	2.31 ± 0.13 ^B^	1.99–2.54
P^5+^ (mmol/L)	1.64 ± 0.56 ^b^	0.49–3.18	1.92 ± 0.57 ^a^	1.25–3.46
Mg^2+^ (mmol/L)	0.9543 ± 0.0975	0.77–1.21	0.9439 ± 0.0950	0.77–1.14
HCY (μmol/L)	6.18 ± 2.87	0.6–13.6	5.97 ± 2.90	2–12.4
CK (U/L)	143.03 ± 61.02	54–375	147.96 ± 57.41	95–340
CKMB (U/L)	115.25 ± 38.06	54–207	110.68 ± 30.68	58–187
LDH (U/L)	1489.12 ± 276.37	803–2331	1540.54 ± 275.62	1177–2218
HBDH (U/L)	1381.17 ± 262.94	754–2185	1436.64 ± 265.32	1103–2085

Note: Values without the same superscript letters in the same row differ at *p* ≤ 0.05 for lower-case and *p* ≤ 0.01 for upper case.

**Table 4 animals-12-01813-t004:** The blood biochemical indicators of Brahman and Yunling cattle.

Item	Yunling Cattle		Brahman		*p*-Value
	Mean ± SD	*n*	Mean ± SD	*n*	
HDL-C (mmol/L)	2.14 ± 0.37 ^B^	65	2.37 ± 0.39 ^A^	28	0.010
TBIL (μmol/L)	2.85 ± 0.99 ^a^	65	2.44 ± 0.74 ^b^	28	0.039
IBIL (μmol/L)	1.74 ± 0.75 ^A^	65	1.33 ± 0.50 ^B^	28	0.002
ALB (g/L)	36.43 ± 2.01 ^A^	65	33.63 ± 3.15 ^B^	28	0.000
ALB:GLO	0.84 ± 0.16 ^a^	65	0.76 ± 0.19 ^b^	28	0.029
GGT (U/L)	19.51 ± 9.42 ^B^	65	32.04 ± 15.82 ^A^	28	0.000
ALP (U/L)	90.97 ± 43.40 ^b^	63	114.43 ± 39.55 ^a^	28	0.017
GLU (mmol/L)	4.28 ± 0.49 ^A^	65	3.80 ± 0.82 ^B^	28	0.003
URIC (μmol/L)	39.86 ± 14.25 ^b^	65	47.75 ± 13.81 ^a^	28	0.015
Cl^−^ (mmol/L)	98.32 ± 3.21 ^b^	65	100.06 ± 3.94 ^a^	28	0.027
Ca^2+^ (mmol/L)	2.38 ± 0.11 ^A^	65	2.31 ± 0.13 ^B^	28	0.006
P^5+^ (mmol/L)	1.64 ± 0.56 ^b^	65	1.92 ± 0.57 ^a^	28	0.028

Note: Values without common superscript letters in the same row differ at *p* ≤ 0.05 for lower-case and *p* ≤ 0.01 for upper case.

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
