# Peer review of "Comparisons of Hematological and Biochemical Profiles in Brahman and Yunling Cattle"

_animals, 2022, doi:10.3390/ani12141813_

Round 1

Reviewer 1 Report

Dear Autors,

the article still needs some corrections to be published.

The title should be changed due to inaccuracy of the methodology.

In the methodology, the mean temperature during the study period was 15 ° C. This temperature is in the comfort zone for cattle. There is no precise information how the temperature data was collected.

The data in the tables is partially duplicated: (data from table 1 in table 2 and data from table 3 in table 4).

Some parameters have high SD (PLT ,PAB, ALP, TBA). How did you test for normality of data distribution?

ALT activity is not specific for hepatic disfunction in cattle.

Author Response

1.Reviewer: 1

Comment 1:

The title should be changed due to inaccuracy of the methodology

Response 1:

The title is changed to “Comparisons of Hematological and Biochemical Profiles in Brahman with Yunling Cattle”

Comment 2:

In the methodology, the mean temperature during the study period was 15 ° C. This temperature is in the comfort zone for cattle. There is no precise information how the temperature data was collected.

Response 2:

The mean temperature was collected by Government of Guandu District, Kunming, China(http://www.kmgd.gov.cn/c/2021-10-18/5450304.shtml). We ignored the comfort temperature for cattle. We will modify the discussion and conclusion of manuscript.

Comment 3:

The data in the tables is partially duplicated: (data from table 1 in table 2 and data from table 3 in table 4).

Response 3:

Thanks for your comment. Table 1 and table 2 are parameters which have significant differences, table 3 and table 4 are all parameters.

Reviewer 2 Report

The article can be accepted as a short communication

Author Response

Thanks for your comment.

Reviewer 3 Report

The presented manuscript “Yunling cattle are more superior than Brahman cattle in physiological status for high temperature and high altitude environment” examines hematological and biochemical parameters in the breeds Yunling and Brahman. The authors draw the conclusion that Yunling cattle are better suited to the environment they were tested in based on these test results. There are some flaws in the design and conclusions drawn from these data as outlined below. Overall, the data could be quite useful in establishing reference values for Yunling cattle if those do not exist yet, with the caveat that they were established in a high altitude, hot environment. Some conclusions could then be drawn about the causes or consequences of differences between the two breeds. The data could be compared to established reference intervals and new reference itervals could be established for these breeds if a certain percentage falls outside of the established reference intervals. The way the data are presented here seems to overinterpret them. The discussion part is lacking specific references to other cattle studies and how similar findings were interpreted there. The references used are mostly based on human studies or other species.

There are no reference values given for any of the parameters under study for either cattle in general, or for these specific breeds. It is hard to tell whether the values found can be considered normal and whether a higher or lower value for any of the parameters represents an advantage or an abnormality.

Statistical analysis: The distribution of variables should be assessed for normality with histograms and non-normally distributed data should be transformed for example, natural log transformation. Otherwise you are risking that outliers will skew your results. In all likelihood, some of the data is not normally distributed. See the attached publication on approved recommendations on the theory of reference values and determination of reference limits.

Specifically:

Line 183: You are speculating that higher levels of white blood cells are linked to greater anti-inflammatory abilities. Do you have any references for this? It may be the other way around, that a higher WBC count could be a sign of more inflammation. A test of activity for the enzymes superoxide dismutase, catalase or glutathione peroxidase would be more helpful in showing ability to fight inflammation in these cattle.

Line 192: The higher RBC count in Yunling cattle could also be due to a lower parasite burden or higher parasite tolerance. A better test to check for high altitude adaptability, could have been to check oxygen saturation via pulse oximetry. In fact, if oxygen saturation was the same then a lower RBC count would indicate better adaptation to high altitude.

Line 215: I am not familiar with the term snowflake beef. In terms of lipids, I am not aware that hyperlipidemia is a cause of concern in cattle in terms of arterial problems. On the other hand, there is evidence that hyperlipidemia can hasten the onset of postpartum luteal activity, see Wehrmann et al. 1991. Biology of Reproduction 45, 514-522: Diet-Induced Hyperlipidemia in cattle modifies the intrafollicular cholesterol environment, modulates ovarian follicular dynamics, and hastens the onset of postpartum luteal activity

Author Response

Comment 1:

The data could be compared to established reference intervals and new reference intervals could be established for these breeds if a certain percentage falls outside of the established reference intervals. The way the data are presented here seems to overinterpret them. The discussion part is lacking specific references to other cattle studies and how similar findings were interpreted there. The references used are mostly based on human studies or other species.

There are no reference values given for any of the parameters under study for either cattle in general, or for these specific breeds. It is hard to tell whether the values found can be considered normal and whether a higher or lower value for any of the parameters represents an advantage or an abnormality.

Response 1:

The discussion part of other cattle studies references is seldom, and we have not found reference values about cattle, so the references used are mostly based on human studies or other species. In the Zhang Jicai’s[1] study, some different cattle’s items are compared, but no reference values.

Comment 2:

Statistical analysis: The distribution of variables should be assessed for normality with histograms and non-normally distributed data should be transformed for example, natural log transformation. Otherwise you are risking that outliers will skew your results. In all likelihood, some of the data is not normally distributed. See the attached publication on approved recommendations on the theory of reference values and determination of reference limits.

Response 2:

Thanks for your comment, we have added the histograms to show the distribution of variables in the draft.

Comment 3:

Specifically:

Line 183: You are speculating that higher levels of white blood cells are linked to greater anti-inflammatory abilities. Do you have any references for this? It may be the other way around, that a higher WBC count could be a sign of more inflammation. A test of activity for the enzymes superoxide dismutase, catalase or glutathione peroxidase would be more helpful in showing ability to fight inflammation in these cattle.

Line 192: The higher RBC count in Yunling cattle could also be due to a lower parasite burden or higher parasite tolerance. A better test to check for high altitude adaptability, could have been to check oxygen saturation via pulse oximetry. In fact, if oxygen saturation was the same then a lower RBC count would indicate better adaptation to high altitude.

Line 215: I am not familiar with the term snowflake beef. In terms of lipids, I am not aware that hyperlipidemia is a cause of concern in cattle in terms of arterial problems. On the other hand, there is evidence that hyperlipidemia can hasten the onset of postpartum luteal activity, see Wehrmann et al. 1991. Biology of Reproduction 45, 514-522: Diet-Induced Hyperlipidemia in cattle modifies the intrafollicular cholesterol environment, modulates ovarian follicular dynamics, and hastens the onset of postpartum luteal activity

Response 3:

Thanks for your comments, we neglected in these respects. We have modified the content.

The snowflake beef is marbled beef, whose fat is distributed evenly and densely in the muscle, like snowflakes or marbling.

Language and Content modifications:

(1) We changed the title from “Yunling cattle are more superior than Brahman cattle in physiological status for high temperature and high altitude environment” to “Comparisons of Hematological and Biochemical Profiles in Brahman with Yunling Cattle”

(2) We screened the data for normal distribution and added the histograms in the supplemental information. Some data were screened, 55 not 56 hematological and biochemical parameters were useful, percentage of low fluorescent reticulum (LRF%) was deleted. Adapting to the humidity-hot stress was changed to local environment, which is more suitable.

(3) We have updated the form.

(4) “The distribution of variables was assessed for normality with histograms in supple-mental information.” Was added in “2.4. Statistical analyses”, P3 L119.

(5) “The Cattle is sensitive to high temperatures, so Cattle bred in southern regions in China, such as Kunming, need to be physiologically adapted to the high temperature environment.” was deleted in P8 L174.

(6) “The total number of RBC and HGB in Yunling cattle were significantly increased, indi-cating that the ability of Yunling cattle to resist heat stress was improved.” was deleted in P8 L192.

(7) “These indicate that Yunling cattle have excellent resistance to heat stress, which not only inherit the strong adaptability of Yunnan cattle, but also inherit the excellent heat resistance of Brahman cattle.” was deleted in P8 L194.

(8) Some language issues were recorrected in the manuscript. Other changes were highlight in the manuscript.

Thanks again for your review and comments, please contact me if you have any questions.

Round 2

Reviewer 3 Report

Thank you for your revised draft of the manuscript. You have made some important modifications, but I still believe the manuscript would benefit from more cattle-specific references on the blood and serum parameters under evaluation. There are certainly hundreds of publications that evaluate cattle blood parameters for different breeds or under different conditions. There are also references that have established reference values for cattle blood parameters, otherwise we wouldn’t know what was considered normal. For example, George JW, Snipes J, Lane M: comparison of bovine hematology reference intervals from 1957 -2006. Vet clin Pathol 39: 138, 2010; Jain NC: Schalm’s veterinary hematology, ed 4, Philadelphia, 1986, Lea & Febiger for hematology or Kaneko JJ, Harvey JW, Bruss ML: Clinical biochemistry of domestic animals, ed 6, Burlingotn MA, 1997, Academic Press; A collection of reference intervals can also be found in Smith BP, Van Metre DC, Pusterla N: Large Animal Internal Medicine, Elsevier

All tables: In the latest version you modified the numbers by adding some decimal places to the results. In general, the mean and standard deviation should be rounded to one more decimal place than what the raw data have. Please adjust for all tables.

Table 2: The note says “Values with no common superscript…” but there are no superscripts – please delete (same for table 4). There is one parameter P-LCR that has a p-value of 0.42. There is no SD, although the header indicates that it is the Mean +/- SD

Please change the first column so that each item is on one line

Line 181: WBC and Mono % and # are not mentioned in table 2 or have differing superscripts in table 1. Are they different?

Since most leukocytes are neutrophils and lymphocytes, what do you think is the significance of differences in the number and % of basophils indicates? I am not convinced that it means the Yunling cattle have better anti-inflammatory ability. Since there are so few basophils in the blood, any differences between the breeds could be by chance or due to measuring error.

Line 192: I would assume that higher HGB and HCT levels may be an indicator or better adaptability to high altitude rather than different climates, where there is less oxygen in the air.

Line 193 – 199: Is there any indication in the literature that a higher platelet count is better for coagulation since hemostasis is a complex process that includes not only platelets but also a number of coagulations factors?

Table 4: ALP: n and values are mixed up

Line 148: the P-value for T-BIL and I-BIL are different from table 4.

Line 151: please avoid a p-value of 0, replace with <0.001 or similar.

Paragraph 3.2: this paragraph is tedious to read and essentially summarizes tables 3 and 4 without adding any additional information. It could be summarized much better by stating which parameters were not different, which were higher or lower in Yunling cattle versus Brahman instead of going through each item one by one.

Line 200 – 202: this statement clearly refers to humans not cattle. I don’t understand the significance in the context of this manuscript.

Line 202: I don’t understand the meaning of this sentence. Please delete or explain what you mean here.

Line 204: The relation of marbling (preferred term to snowflake) and serum cholesterol is interesting but again, you are not citing any cattle studies. For example, there is a study: Matsuzaki M.Takizawa S. Ogawa M. 1997 . Plasma insulin, metabolite concentrates, and carcass characteristics of Japanese Black, Japanese Brown, and Holstein steers. J. Anim. Sci. 75 :3287–3293 that should be included here.

Line 221: “Liver function damage generally does not show obvious clinical symptoms: “ Liver function damage can show as jaundice, photosensitivity, or hepatic encephalopathy, so I would delete this statement.

Line 236: Again, the reference to human diseases such as alcoholic liver or liver cancer is of little value for this study. Please look up what GGT levels represent in cattle.

Line 244: Bailey’s study is Nr. 29 in the list – other references may need to be renumbered as well.

Line 245: I would really refrain from making such anthropomorphisms.

Line 261: HCO3- is a buffer to maintain the body’s pH level. The term “homeostasis balance” is awkward.

Line 264: What are you trying to say with this sentence? Were there large differences between individuals? You think that some of them are in a state of dysfunction?

Line 267: CK is mainly an indicator of muscle damage, where modest elevations are fourfold increases over resting values. The minimal difference between the breeds you observed (not sig. as far as I can tell) may have little meaning.

Conclusions:

After re-reading the manuscript, there is little evidence for greater anti-inflammatory and skeletal and heart muscle function, so I would delete those. The other traits are also speculative and must be confirmed with actual studies testing these hypotheses specifically. You need to include a statement to this effect in the conclusions.

Round 3

Reviewer 3 Report

Thank you for incorporating further edits to your manuscript. In order to improve readability, I strongly recommend having the text reviewed by an English speaking copy editor. I believe the publisher offers such services for a fee.

This manuscript is a resubmission of an earlier submission. The following is a list of the peer review reports and author responses from that submission.

Round 1

Reviewer 1 Report

Thank you for the opportunity to review the manuscript entitled “Comparison of hematological and biochemical profiles in Brahman with Yunling Cattle”. Unfortunately, I will have to suggest the rejection of the manuscript. The Introduction section does not provide the relevant background and the objectives are not clearly stated. The Materials and Methods section is not accurate and thorough. The presentation of the results is suboptimal. The Discussion section is not focused on the findings of the study. I cannot provide any specific comments as the manuscript needs to be re-written or fundamental amendments need to be done at a minimum. The English language needs to be improved essentially.

Reviewer 2 Report

The article needs minor revision and can be considered as short communication.

There are few syntax errors and grammatical mistakes in manuscript. Pay special attention to the clarity of the message, length and structure of the sentence. 

Mention the details of Automatic Biochemical Analyzer

Reviewer 3 Report

The manuscript entitled "Comparisons of Hematological and Biochemical Profiles in Brahman with Yunling Cattle" presents comeration of hematological and biochemical parameters in two cattle breeds, an interesting and important topic. However, the novelty of the study is limited.